# Current Epidemiology of the General Anesthesia Practice for Cesarean Delivery Using a Nationwide Claims Database in Japan: A Descriptive Study

**DOI:** 10.3390/jcm11164808

**Published:** 2022-08-17

**Authors:** Hiroshi Yonekura, Yusuke Mazda, Shohei Noguchi, Hironaka Tsunobuchi, Motomu Shimaoka

**Affiliations:** 1Department of Molecular Pathobiology and Cell Adhesion Biology, Mie University Graduate School of Medicine, 2-174 Edobashi, Tsu 514-8507, Japan; 2Department of Anesthesiology and Pain Medicine, Fujita Health University Bantane Hospital, 3-6-10 Otoubashi, Nakagawa-ku, Nagoya 454-8509, Japan; 3Department of Obstetric Anesthesiology, Center for Maternal-Fetal and Neonatal Medicine, Saitama Medical Center, Saitama Medical University, 1981 Kamoda, Kawagoe 350-8550, Japan

**Keywords:** administrative database, cesarean delivery, general anesthesia, obstetric anesthesia

## Abstract

The current status of general anesthesia practice for cesarean delivery in Japan remains unknown. Therefore, using a nationwide claims database, we aimed to investigate general anesthesia use for cesarean delivery over a period of 15 years, and to analyze the general anesthesia practice in Japan. Patients who claimed the Japanese general anesthesia claim code (L008) for cesarean delivery between 1 January 2005, and 31 March 2020, were analyzed. Primary endpoint was the prevalence of general anesthesia use. We used two definitions of general anesthesia: L008 code only (insurance definition) and combination of the L008 code with muscle relaxant use (clinical definition). The general anesthesia claim cohort (L008) included 10,972 cesarean deliveries at 1111 institutions from 2005 to 2020. Muscle relaxants were used in 27.3% of L008 claims cases. The rate of general anesthesia use for cesarean delivery ranged from 3.9% in clinical definition to 14.4% in insurance definition of all cesarean deliveries. We observed a temporal trend of gradual decrease in general anesthesia use, regardless of its definition (*p* for trend < 0.001). We recommend the clinical definition of general anesthesia as the combination of L008 code and muscle relaxant use in a claims-based approach.

## 1. Introduction

In the Japanese perinatal care delivery system, there are over 2500 medical facilities for approximately 1 million deliveries per year, of which 18.6% were cesarean deliveries [1,2]. Compared to Western countries, the Japanese delivery system is characterized by small-scale facilities that handle deliveries in a decentralized manner with localized networks. According to the Japanese 2017 vital statistics of the Ministry of Health, Labour, and Welfare, deliveries at hospitals and clinics (i.e., facilities with <20 beds) account for 54.3% and 45.7% of all deliveries, respectively. Considering that approximately 31% of all cesarean deliveries are performed in clinics [3], obstetric research focusing on only hospital-level facilities does not accurately reflect the current state of the clinical practice of cesarean delivery.

The modes of anesthesia for cesarean delivery can be classified into two main categories: general and neuraxial anesthesia. Given the risks of difficult tracheal intubation, hypoxemia, aspiration, and accidental awareness, general anesthesia should be avoided in cesarean deliveries unless clinically indicated [4,5]. In the United States (US), the rate of general anesthesia use between 2010 and 2015 was 5.8% for all cesarean deliveries and 14.6% for emergency cases [6]. As there is a limited database that could be used to investigate the current state of the clinical practice of cesarean delivery, only a few studies address the rate of general anesthesia use in Japan. To date, therefore, the current status of general anesthesia use across Japanese medical facilities in real-world settings remains unclear.

In the Japanese insurance system, the definition of general anesthesia can be problematic. General anesthesia is coded as L008: “general anesthesia used in a closed-circuit system maintained by mask or mechanical ventilation” (Appendix A). This definition does not necessarily require airway-securing devices; the definition of general anesthesia includes the administration of a general anesthesia agent for ≥20 min in combination with oxygen or oxygen/nitrous oxide mixed gas by mask or endotracheal intubation. For example, neuraxial anesthesia administered in combination with intravenous sedation and oxygen supplementation via mask may be misclassified and claimed as general anesthesia, depending on an institutional or organizational policy.

The aim of this study was to investigate general anesthesia use for cesarean delivery over 15 years and to analyze the general anesthesia practice in Japan.

## 2. Materials and Methods

### 2.1. Data Source

This retrospective cohort study was approved by the Institutional Review Boards of Mie University and Fujita Health University (study approval numbers: H2019-167 and HM21-369, respectively). The requirement for additional informed consent from the patients was waived owing to the anonymous nature of the data. This study was conducted according to the Strengthening the Reporting of Observational Studies in Epidemiology Statement [7].

The medical claims data for this study were obtained from a medical database vendor (JMDC Co. Ltd., Tokyo, Japan) [8]. The JMDC database accumulates the reimbursement data of over 100 million insured persons from 2005, which represent approximately 10% of the Japanese population. This insurance-based database is suitable for maternal healthcare research because it collects claims data from over 250 health insurance associations whose enrollees are employees of medium-to-large companies and their family members. The JMDC database can provide comprehensive tracking data of outpatient and inpatient maternal patients chronologically, regardless of whether maternal patients visit the emergency department or require hospitalization. The database contains the following information: patient demographic data (e.g., age and sex), inpatient and outpatient claims data (e.g., medical and pharmacy claims for diagnosis, procedure, and medication), and facility information. Clinical diagnoses are coded according to the International Classification of Diseases 10th revision (ICD-10), procedural information is defined using Japanese standardized procedure codes (K codes), and medication information is coded according to the World Health Organization Anatomical Therapeutic Chemical (WHO-ATC) classification system [9]. The quality of the data collected within the Japanese administrative data is generally guaranteed [10].

### 2.2. Study Population

Using the original Japanese K codes, we identified cesarean deliveries performed between 1 January 2005, and 31 March 2020, as follows: K898-2 for elective cesarean delivery, K898-1 for emergency cesarean delivery, and K898-3 (this procedure code was only used between 2010 and 2016) for cesarean delivery with placenta previa or preterm birth (i.e., birth before 32 weeks of gestation) [11]. Cases with missing anesthesia records were excluded from this study. Of the overall cesarean delivery cohort (general and neuraxial anesthesia cases), only cesarean deliveries with general anesthesia claim (L008) were analyzed (general anesthesia claim cohort).

### 2.3. Patient- and Facility-Level Variables

We assessed the following patient characteristics: age, Maternal Comorbidity Index (MCI), Charlson Comorbidity Index (CCI), and obstetrical/maternal/fetal risk factors associated with increased likelihood of general anesthesia use based on a previous study [12] (e.g., amniotic fluid embolism, chorioamnionitis, fetal distress, obstetric hemorrhage, placental abruption, placenta accreta, placental dysfunction, umbilical cord prolapse, and uterine rupture) (Appendix A). To determine the presence of specific comorbidities, we used the diagnostic ICD-10 codes, MCI developed by Bateman et al. [13], and CCI [14]. MCI is a validated scoring system for obstetric populations that is designed to assess maternal comorbidity [15,16].

Data on medical facilities included the number of beds (0–19, 20–99, 100–199, 200–299, 300–499, or >499 beds), facility characteristics, and payment system. We classified the facilities according to their characteristics into academic hospitals and non-academic hospitals [9]. We used the Diagnosis Procedure Combination (DPC) payment system as a surrogate for acute care hospitals.

### 2.4. Outcomes

The primary endpoint was the prevalence of general anesthesia use. We used the following two definitions of general anesthesia: (1) insurance definition and (2) clinical definition. In general, general anesthesia was coded as L008 in the database (“insurance definition”). However, given that, in the Japanese health insurance system, the clinical anesthesia definition differs from the insurance definition, we developed second definitions of general anesthesia by combining the Japanese L008 codes and WHO-ATC drug claims. Rapid sequence induction (RSI) with securing of the airway is the standard induction technique for cesarean delivery under general anesthesia [5]. A previous study has revealed that RSI was the preferred induction method for women undergoing obstetric surgery in England, and neuromuscular blocking agents (NMBAs) were used in 98% of cases [17]. Therefore, as “clinical definition” of general anesthesia, we defined general anesthesia as the combination of L008 and use of an NMBA (WHO-ATC drug code: M03) to avoid misclassification, regardless of whether regional block was performed.

### 2.5. Statistical Analysis

Patient, medical facility, and medication characteristics were assessed for cases with and without NMBAs. Continuous variables were presented as mean and standard deviation (SD) or as median and interquartile range (IQR), and categorical variables were presented as numbers and percentages. In general, the larger the sample size, the smaller the *p*-value obtained from the data for the baseline comparison. Therefore, we calculated the standardized mean difference (SMDs) between L008 cases with and without NMBAs; an absolute SMD > 10% indicates meaningful imbalance between group differences [18]. We calculated the proportions of general anesthesia use for the entire cohort and described the trend of general anesthesia practice for cesarean deliveries from 2005 to 2020. The rate of general anesthesia use was calculated yearly and stratified according to the general anesthesia definition (i.e., insurance and clinical anesthesia definitions). The trends of proportions were assessed using the Cochran–Armitage trend test [19].

All analyses were performed using SAS 9.4 for Windows (SAS Institute Inc., Cary, NC, USA).

## 3. Results

### 3.1. Study Cohort

Figure 1 shows the flow diagram. In the initial cohort of patients, 77,640 cesarean deliveries were considered eligible procedures; of these, we excluded 1460 procedures owing to missing anesthesia information. Thus, an overall cohort of 76,180 procedures involving 64,761 women at 2506 institutions between 1 January 2005, and 31 March 2020, were included in the analysis. After the exclusion of neuraxial anesthesia cases (85.6%, *n* = 65,208), the general anesthesia claim cohort defined by the L008 code included 10,108 patients who underwent 10,972 cesarean deliveries at 1111 institutions. Of the general anesthesia claim cohort (L008), the mean patient age (standard deviation) was 33.2 (5.0) years, there were 4301 emergency operations (39.2%), and 41.6% of the cesarean deliveries were performed in facilities with <20 beds. NMBAs were used in only 27.3% of the general anesthesia claim cohort (*n* = 2992).

### 3.2. Patient and Medical Facility Characteristics

Table 1 summarizes the patient demographic and facility characteristics of the study population who did and did not receive NMBAs. The group who received NMBAs was more likely to have obstetrical comorbidities and a higher burden than the group who did not receive NMBAs. The obstetrical/maternal/fetal risk factors were more likely to occur in the group who received NMBAs than in those who did not. The group who received NMBAs was more likely to undergo emergency cesarean deliveries in a larger medical facility, especially >499 bed size, academic hospitals, and DPC payment system hospitals (Table 1).

Table 2 presents a comparison of intraoperative drugs used in the study cohort with general anesthesia claim code (L008) according to the use of NMBAs. The NMBAs used for RSI were rocuronium (in 78.5% of cases) and suxamethonium (in 33.9% of cases). The use of vecuronium and pancuronium is not recommended for RSI for cesarean delivery owing to the slow onset of action; therefore, these drugs were probably used in an addition to the fast-acting suxamethonium. Regarding general anesthetics, the group that received NMBAs was more likely to receive propofol (62.0% vs. 42.9%), barbiturate (47.5% vs. 4.1%), and volatile halogenated agents (67.8% vs. 15.1%) compared to the group that did not receive NMBAs. In contrast, benzodiazepine, nitrous oxide, and ketamine were more likely used in the group that did not receive NMBAs. The group that received NMBAs was also more likely to receive fentanyl (68.2% vs. 37.0%) and remifentanil (48.4% vs. 0.2%). Regarding the group that did not receive NMBA (7980 cases), spinal anesthesia was used in 63.9% of cases.

### 3.3. Trends in the Rate of General Anesthesia Use

Figure 2 shows the longitudinal changes in general anesthesia use from 2005 to 2020 according to the general anesthesia definition (insurance and clinical definitions). The rates of general anesthesia use in the insurance definition in 2005, 2010, 2015, and 2020 were 29.5%, 22.5%, 12.9%, and 11.5%, respectively (*p* for trend < 0.001). The temporal trend of the clinical definition of general anesthesia also decreased gradually from 9.9% in 2005 and 4.7% in 2010 to 3.3% in 2020 (*p* for trend < 0.001) (Figure 2 and Appendix A).

Regarding medical facilities, of all L008 claim cases, 41.6% of the cases (4567/10,972) were claimed in obstetric facilities with <20 beds. Furthermore, 88.6% (100–11.4) of L008 cases at obstetric facilities with <20 beds did not use NMBAs (Table 1 and Figure 3A). Figure 3B shows that the percentage of L008 cases with and without NMBAs is proportional to the facility’s bed size. The L008 code with NMBAs was claimed for 67.3% of cases in large facilities with >499 beds and 11.4% of cases in facilities with <20 beds.

## 4. Discussion

In this study, we identified the current epidemiology of general anesthesia for cesarean delivery in real-world setting in Japan and the inherent problems of these reimbursements. This information is important to understand, guide future study, and inform not only physicians but also policymakers in Japan. Our results showed a temporal trend of gradual decrease in general anesthesia use, regardless of its definition. The majority of L008 cases at obstetric facilities with <20 beds did not receive NMBAs. Our study strongly suggests that 72.7% of general anesthesia cases with the insurance definition cannot be clinically defined as general anesthesia.

Our analysis revealed that the rate of general anesthesia use for cesarean delivery ranged from 3.9% in clinical definition to 14.4% in insurance definition. The rate of general anesthesia use can be used as a quality indicator to support the monitoring of local performance and its improvement. In the quality improvement compendium, the Royal College of Anaesthetists raised the standards of quality indicators of care to >95% for elective cesarean deliveries performed under neuraxial anesthesia and to >85% for emergency cases [20]. Our study demonstrated that the temporal trend of general anesthesia with the clinical definition decreased gradually from 9.9% in 2005 to 3.3% in 2020, which is consistent with the finding of a US study that revealed a national trend of decrease in the rate of general anesthesia use from 13.2% in 2010 to 2.6% in 2015 [6]. A previous survey of a 30-year obstetric anesthesia workforce in the US also revealed a constant low rate of elective general anesthesia use but a slight increase in the rate of emergency cases [21]. Czech and Slovak national surveys revealed an increase in the use of neuraxial anesthesia, but the proportion of general anesthesia remained high (24%) [22]. Our data can act as a benchmark when international data are compared to current obstetric anesthesia practice in Japan.

In the study by Abe et al., general anesthesia was used in 11.3% of elective cesarean deliveries from 2010 to 2013 [11]; this rate is significantly higher than in our study’s general anesthesia cases with the clinical definition and in the cases of previous studies conducted in high-income countries [23]. The high rate of general anesthesia use in the study by Abe et al. is explained partially by the characteristics of the database used in their study, especially as the database mainly contains data from acute care hospitals and high-volume centers in Japan (in the setting of DPC hospitals). In a retrospective review of 10-year trends in general anesthesia practice for cesarean delivery at a university hospital (i.e., DPC hospitals) in Japan, they reported a steady decline in the percentage of elective cesarean deliveries performed under general anesthesia from 11.1% in 2010 to 4.4% in 2019 [24]. However, 31% of cesarean deliveries are also performed in small facilities in Japan [3]; thus, a study that focuses only on hospital-level facilities does not accurately reflect current obstetrics anesthesia practice in Japan. Moreover, Japanese anesthesia-related codes (L code) are the basis for the determination of the type of anesthesia; therefore, coding accuracy or miscoding can lead to misclassification bias. Considering the difficulty associated with identifying general anesthesia using only anesthesia claims codes, the problem of reimbursement may explain the overestimation of the rate of general anesthesia use in the previous Japanese study on cesarean delivery [11]. In fact, our study results show that regarding the L008 cases without NMBAs, the administration of general anesthetics, such as propofol, barbiturate, volatile agents, and remifentanil, was disproportionately low compared to the L008 cases with NMBAs (clinical definition of general anesthesia). This reimbursement problem may explain the overestimation of the rate of general anesthesia use in the study [11]. Our data strongly suggest that 72.7% of the general anesthesia cases with the insurance definition cannot be clinically defined as general anesthesia. We recommend the clinical definition of general anesthesia as the combination of L008 code and the use of NMBAs in a claimed-based approach.

Among the general anesthesia claim cohort (L008), 41.6% were claimed in obstetric facilities with <20 beds. Small facilities do not meet the standards of regular operating rooms as proposed by the European minimum standards and WHO [25,26]. Small facilities do not have sufficient equipment (operating rooms), resources (drug availability), and work forces (anesthesiologists and neonatologists). Resuscitation may be difficult without adequate in-house neonatologists when an anesthetized baby is delivered after general anesthesia. Therefore, our study revealed potential barriers to improving maternal and neonatal safety and standardizing obstetrical anesthesia practice.

We cannot investigate why the majority (88.6%) of cesarean deliveries at small facilities (also known as “clinics”) did not use NMBAs, but they requested reimbursement in general anesthesia claims (L008) in our database. One possible explanation is that the “general anesthesia” itself in the health insurance system is vaguely defined. In terms of insurance reimbursement, general anesthesia reimbursement is 7 times higher than that of neuraxial anesthesia (Appendix A), which may lead to incentives for small facilities, especially private obstetrical clinics.

The strength of our study is that it is the largest nationwide study to include diverse facilities and reflects the current state of clinical practice to date. The JMDC database is limited to the employee-based insured population; however, the Japanese health care system has universal health coverage. Hence, different types of health insurance or racial/social-economic disparities would not influence our results. Thus, our findings are likely indicative of the majority of cesarean deliveries performed in Japan [9].

The limitations of this study include the use of a claims-based database, which does not include indications for general anesthesia and potential confounding factors. Our claims-based database does not distinguish between the conversion from failed neuraxial anesthesia to general anesthesia and the combined use of general and neuraxial anesthesia. Cases of L008 with NMBAs could include conversion or concomitant use; however, it is impossible to distinguish between the two retrospectively. Second, coding inconsistencies or misclassification can affect our definition of general anesthesia. To avoid misclassification, we developed the clinical definition of general anesthesia as the combination of L008 code and the use of NMBAs. In a recent study, NMBAs were administered in more than 98% of cases of general anesthesia in women undergoing obstetric surgery in England [17]. Future validation studies of our definition of general anesthesia are warranted.

## 5. Conclusions

Identifying the current epidemiology of general anesthesia for cesarean delivery and inherent problems of reimbursements in Japan is important to understand, guide future study, and inform not only physicians but also policymakers. Our analysis revealed that the rate of general anesthesia use for cesarean delivery ranged from 3.9% to 14.4% of all cesarean deliveries depending on the definition. NMBAs were used in only 27.3% of L008 cases. We recommend the clinical definition of general anesthesia as the combination of L008 code and the use of NMBAs in a claimed-based approach. Future validation studies of our claims-based approach to identify the general anesthesia practice are warranted.

## Figures and Tables

**Figure 1 jcm-11-04808-f001:**
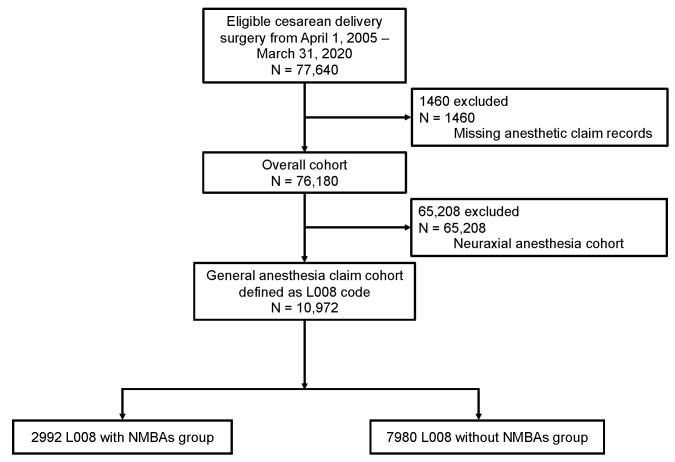
Flow diagram.

**Figure 2 jcm-11-04808-f002:**
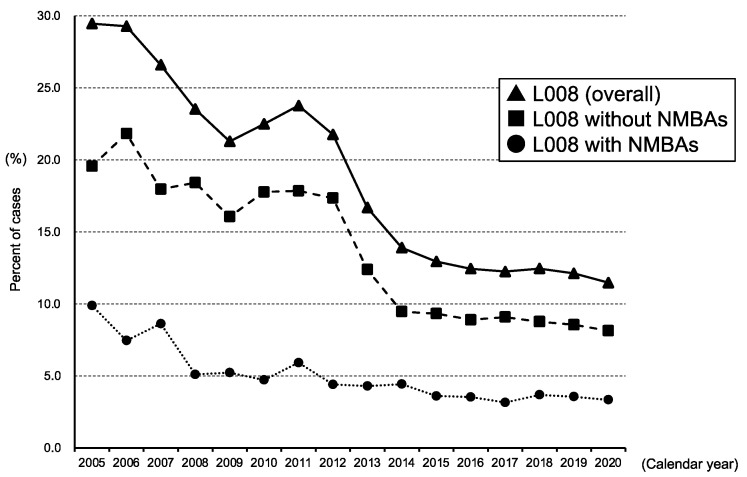
Annual rates of general anesthesia claim code (L008) with and without neuromuscular blocking agents (NMBAs) in total cesarean delivery cases from 2005 to 2020.

**Figure 3 jcm-11-04808-f003:**
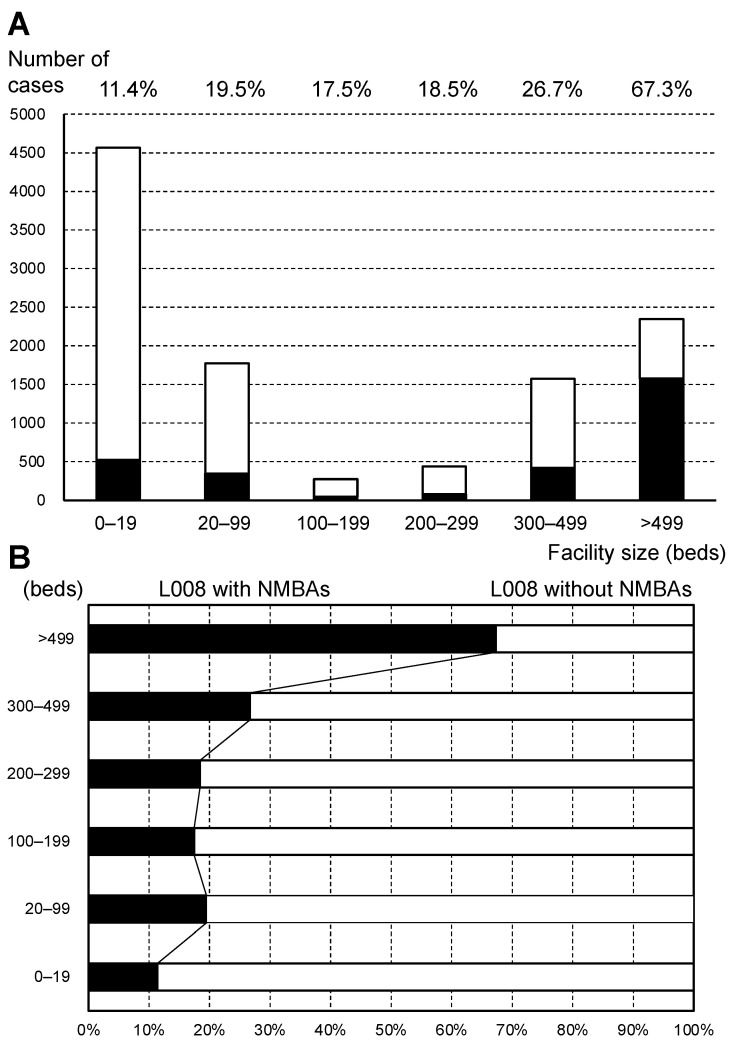
(**A**) The number of general anesthesia claim code (L008) according to facilities with different bed sizes. The black bar indicates the number of cesarean deliveries receiving NMBAs during general anesthesia. (**B**) The percentage of L008 with and without neuromuscular blocking agents (NMBAs) according to the facilities with different bed sizes.

**Table 1 jcm-11-04808-t001:** Patient and medical facility characteristics of the study cohort with general anesthesia claim code (L008) according to the use of neuromuscular blocking agents (NMBAs).

Characteristic	L008 with NMBA, *n* = 2992	L008 without NMBA, *n* = 7980	Absolute SMD (%)
Age, year			5.0
Mean ± SD	33.4 ± 5.1	33.1 ± 4.9	
<35	1826 (61.0)	5060 (63.4)	4.9
35–39	869 (29.0)	2241 (28.1)	2.1
40–44	266 (8.9)	616 (7.7)	4.2
>44	31 (1.0)	63 (0.8)	2.6
Maternal Comorbidity Index			
Alcohol abuse	2 (0.1)	2 (0.0)	1.9
Asthma	241 (8.1)	616 (7.7)	1.2
Cardiac valvular disease	23 (0.8)	16 (0.2)	8.2
Chronic congestive heart failure	5 (0.2)	1 (0.0)	5.2
Chronic ischemic heart disease	32 (1.1)	27 (0.3)	8.8
Chronic renal disease	26 (0.9)	26 (0.3)	7.1
Congenital heart disease	24 (0.8)	15 (0.2)	8.8
Drug abuse	0 (0)	0 (0)	
Gestational hypertension	32 (1.1)	51 (0.6)	4.7
Human immunodeficiency virus	0 (0)	0 (0)	
Mild/unspecified pre-eclampsia	113 (3.8)	246 (3.1)	3.8
Multiple gestation	120 (4.0)	198 (2.5)	8.6
Placenta previa	270 (9.0)	374 (4.7)	17.2
Pre-existing diabetes mellitus	41 (1.4)	40 (0.5)	9.0
Pre-existing hypertension	88 (2.9)	106 (1.3)	11.2
Previous cesarean delivery	193 (6.5)	938 (11.8)	18.5
Pulmonary hypertension	4 (0.1)	0 (0)	5.2
Severe pre-eclampsia	111 (3.7)	246 (3.1)	3.5
Sickle cell disease	0 (0)	0 (0)	
Systemic lupus erythematosus	21 (0.7)	13 (0.2)	8.2
Maternal Comorbidity Index score, median (IQR)	1 (0–2)	1 (0–1)	
0	1298 (43.4)	3767 (47.2)	7.7
1–2	1237 (41.3)	3407 (42.7)	2.7
>2	457 (15.3)	806 (10.1)	15.6
Obstetrical/maternal/fetal emergency			
Amniotic fluid embolism	8 (0.3)	0 (0)	7.3
Chorioamnionitis	287 (9.6)	418 (5.2)	16.7
Fetal distress	675 (22.6)	1260 (15.8)	17.3
Obstetric hemorrhage	8 (0.3)	8 (0.1)	3.9
Placental abruption	375 (12.5)	87 (1.1)	46.6
Placenta accreta	51 (1.7)	27 (0.3)	13.6
Placental dysfunction	353 (11.8)	476 (6.0)	20.6
Umbilical cord prolapse	70 (2.3)	102 (1.3)	8.0
Uterine rupture	20 (0.7)	12 (0.2)	8.1
Type of cesarean delivery			
Elective (K898-2)	1093 (36.5)	5343 (67.0)	63.9
Emergency (K898-1)	1726 (57.7)	2575 (32.3)	52.8
Cesarean delivery with placenta previa or preterm birth (K898-3) *	173 (5.8)	62 (0.8)	28.4
Fiscal year			
2005–2009	231 (7.7)	626 (7.8)	0.5
2010–2014	874 (29.2)	2588 (32.4)	7.0
2015–2020	1887 (63.1)	4766 (59.7)	6.9
Number of beds			
0–19	520 (17.4)	4047 (50.7)	75.1
20–99	346 (11.6)	1429 (17.9)	18.0
100–199	48 (1.6)	227 (2.8)	8.4
200–299	81 (2.7)	358 (4.5)	9.6
300–499	419 (14.0)	1152 (14.4)	1.2
>499	1578 (52.7)	767 (9.6)	105.2
Academic hospital	656 (21.9)	174 (2.2)	63.6
DPC payment system hospital	2109 (70.5)	2380 (29.8)	89.0

Values given as frequencies (%) unless stated otherwise. Absolute SMD > 10% indicates meaningful imbalance between group differences. DPC, Diagnostic Procedure Combination; IQR, interquartile range; NMBAs, neuromuscular blocking agents; SD, standard deviation; SMD, standardized mean difference. * Procedure code only available from 2010 to 2016.

**Table 2 jcm-11-04808-t002:** Comparison of intraoperative drugs used in the study cohort with general anesthesia claim code (L008) according to the use of neuromuscular blocking agents (NMBAs).

	L008 with NMBAs, *n* = 2992	L008 without NMBAs, *n* = 7980	Absolute SMD (%)
NMBAs *	2992 (100)	0 (0)	
Rocuronium	2349 (78.5)	0 (0)	270.3
Suxamethonium	1014 (33.9)	0 (0)	101.2
Vecuronium	298 (10.0)	0 (0)	47.0
Pancuronium	36 (1.2)	0 (0)	15.6
Anesthetics			
Propofol	1855 (62.0)	3425 (42.9)	38.9
Barbiturate	1422 (47.5)	326 (4.1)	114.3
Benzodiazepine	391 (13.1)	1487 (18.6)	15.3
Volatile halogenated agents	2030 (67.8)	1202 (15.1)	126.9
Nitrous oxide	1237 (41.3)	5064 (63.5)	45.4
Ketamine	212 (7.1)	791 (9.9)	10.1
Analgesics			
Fentanyl	2040 (68.2)	2951 (37.0)	65.8
Remifentanil	1447 (48.4)	15 (0.2)	135.8
Morphine (any route)	239 (8.0)	1171 (14.7)	21.2
Local anesthetic for spinal anesthesia	423 (14.1)	5103 (63.9)	118.7
Bupivacaine for spinal anesthesia	410 (13.7)	4637 (58.1)	104.4
Local anesthetic for epidural anesthesia	1032 (34.5)	4463 (55.9)	44.1
2% Lidocaine	85 (2.8)	602 (7.5)	21.3
2% Mepivacaine	65 (2.2)	1154 (14.5)	45.6
Ropivacaine	728 (24.3)	3012 (37.7)	29.3
Levobupivacaine	232 (7.8)	378 (4.7)	12.5
Bupivacaine, excluding for spinal anesthesia	46 (1.5)	881 (11.0)	39.9
Code L003: continuous infusion of local anesthetic after epidural anesthesia	349 (11.7)	3105 (38.9)	66.0

Values given as frequencies (%) unless stated otherwise. Absolute SMD > 10% indicates meaningful imbalance between group differences. NMBAs, neuromuscular blocking agents; SMD, standardized mean difference. * From 2005 to 2020, only rocuronium, vecuronium, pancuronium, and suxamethonium were available for clinical use in Japan.

## Data Availability

The datasets analyzed in this study are available from the corresponding author on reasonable request.

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
