# Peer review of "Current Epidemiology of the General Anesthesia Practice for Cesarean Delivery Using a Nationwide Claims Database in Japan: A Descriptive Study"

_jcm, 2022, doi:10.3390/jcm11164808_

Round 1

Reviewer 1 Report

The authors have presented a 16 year old audit of caesarean deliveries conducted in Japan. The limitations of the manuscript are those which are inherent with a retrospective analysis. The authors have presented their data which is interesting though limited by geographical constraints as this would be of interest to Japanese clinicians and managers. They have compared their findings with the rest of the world and have formulated discussion accordingly.

Author Response

Response to the Reviewer #1

We wish to express our appreciation to Reviewer #1 for providing insightful comments, which have helped us improve our manuscript significantly. We agree with all comments and have incorporated them into the R1 version of the manuscript.

Comment 1: The authors have presented a 16 year old audit of caesarean deliveries conducted in Japan. The limitations of the manuscript are those which are inherent with a retrospective analysis. The authors have presented their data which is interesting though limited by geographical constraints as this would be of interest to Japanese clinicians and managers. They have compared their findings with the rest of the world and have formulated discussion accordingly.

Response: We are grateful for the Reviewer’s appreciation.

Reviewer 2 Report

In the paper authors aimed to investigate general anesthesia use for cesarean delivery over 15 years and to analyze the general anesthesia practice in Japan.

However, they identified a problem with reliable reporting to the insurance nationwide database general anesthesia (GA) for cesarean section (CS) rates resulting, according to the authors' suggestions, from problems with reimbursement for cesarean delivery procedures.

In the manuscript, the authors provide data indicating the vast discrepancy in reporting GA rates in 2005-2020, ranging from 3,9%  for 'clinical' definition (NMBA use) to 14,4% for 'insurance' definition (without NMBA use).

There are many confounding factors like:

·       The use of general anesthetics (propofol, barbiturate, volatile agents, ketamine), i.e., 12 295 procedures in 7980 cases of reported general anesthesia 'insurance' definition, which according to the authors, should not be included in the GA group

·       Use of vecuronium and pancuronium as separate cases of GA (in 'clinical' definition group – NMBAs), which are not recommended for RSI for induction of GA for CS due to slow onset of action. These drugs probably were an addition to short-acting Suxamethonium (?) (for a total of 2992 'clinical' definition GA, there were 3697 reports of different NMBA use), which is not considered or discussed by the authors

·       The authors provide data on neuraxial anesthesia (NA) use in both reported GA groups. Five thousand one hundred three spinal and 4463 epidural local anesthetic use for 'insurance' definition (with a total of  7980 GA vs 9566 NA) and 423 spinal and 1032 epidural local anesthetic use for 'clinical' definition ( with a total of 2992 GA with 1455 NA?) BUT do not provide any explanation of data on conversion from neuraxial to GA rate or CSE use… ???

·       The authors do not provide data on the number of deliveries covered when reporting data for 15 years. They initially reported data from 2 506 units; after excluding missing data records (1460), there are 1 111 units that uploaded records.
It is impossible to find out what was CS rate for this period.

·       Following that, if one considers provided data correct, it gives 10 972 GA of all CS done in 1111 units for 15 years resulting in a performance rate of fewer than 10 CS per 15 years per unit = less than 1 GA cs per unit per year??? 

·       For clinical definition, even less … 2 992 GA, which results in 2,69 GA CS for a unit for 15 years (?!?) = 0,18 of GA CS/unit/year, which is an extremely low number…. And indicates possible underreporting of GA use from very selected groups/units/patient population. Not consecutive cases allow for data generalization for the whole population of Japan…

Minor

Reported data are doubled in Figures and tables (Figure 2 and Table 3)

Incorrect order of tables

The details of the database should be included in the manuscript's content, avoiding unnecessary citations.

Avoid terms like "real-world clinical practice" 

Author Response

Response to the Reviewer #2

We wish to express our appreciation to the Reviewer #2 for providing insightful comments, which have helped us improve our manuscript significantly. We agree with all comments and have incorporated them into the R1 version of the manuscript.

Comment 1: There are many confounding factors like:
The use of general anesthetics (propofol, barbiturate, volatile agents, ketamine), i.e., 12295 procedures in 7980 cases of reported general anesthesia 'insurance' definition, which according to the authors, should not be included in the GA group

Response: We appreciate the Reviewer’s valuable comment. We believe that the insurance definition of "general anesthesia" misclassifies this issue. Under the current Japanese insurance system, cases wherein a sedative (propofol, barbiturate, volatile agents, ketamine) is used in combination with a neuraxial anesthesia may be misclassified as general anesthesia. To resolve this misclassification, we have defined general anesthesia in clinical practice as the concomitant use of muscle relaxants. In a recent study, muscle relaxants were administered in more than 98% of cases of general anesthesia for cesarean deliveries (Anaesthesia 2021, 76, 460–471, doi:10.1111/anae.15250). Therefore, we believe that our clinical definition is more accurate in describing the current status of general anesthesia in Japan.

In response to the reviewers' suggestions, we have added the following text and references.

page 3, lines 123–126:

“Rapid sequence induction (RSI) with securing of the airway is the standard induction technique for cesarean delivery under general anesthesia [5]. A previous study has revealed that RSI was the preferred induction method for women undergoing obstetric surgery in England, and neuromuscular blocking agents (NMBAs) were used in 98% of cases [17].”

page 11, lines 301–303:

“In a recent study, NMBAs were administered in more than 98% of cases of general anesthesia in women undergoing obstetric surgery in England [17].”

[17] Odor, P.M.; Bampoe, S.; Moonesinghe, S.R.; Andrade, J.; Pandit, J.J.; Lucas, D.N.; Pan-London Perioperative Audit and Research Network (PLAN), for the DREAMY Investigators Group General Anaesthetic and Airway Management Practice for Obstetric Surgery in England: A Prospective, Multicentre Observational Study*. Anaesthesia 2021, 76, 460–471, doi:10.1111/anae.15250.

Comment 2: Use of vecuronium and pancuronium as separate cases of GA (in 'clinical' definition group – NMBAs), which are not recommended for RSI for induction of GA for CS due to slow onset of action. These drugs probably were an addition to short-acting Suxamethonium (?) (for a total of 2992 'clinical' definition GA, there were 3697 reports of different NMBA use), which is not considered or discussed by the authors

Response: We appreciate the Reviewer’s valuable comment. From our data, we could not determine whether the muscle relaxants were used for induction or maintenance purposes. As the Reviewer has indicated, we believe that rocuronium and suxamethonium were clinically considered for induction purposes. In accordance with the Reviewer’s comment, we have modified the following text to avoid any misunderstanding (page 6, lines 172–176):

“The NMBAs used for RSI were rocuronium (in 78.5% of cases) and suxamethonium (in 33.9% of cases). The use of vecuronium and pancuronium is not recommended for RSI for cesarean delivery owing to the slow onset of action; therefore, these drugs were probably used in an addition to the fast-acting suxamethonium.”

Comment 3: The authors provide data on neuraxial anesthesia (NA) use in both reported GA groups. Five thousand one hundred three spinal and 4463 epidural local anesthetic use for 'insurance' definition (with a total of 7980 GA vs 9566 NA) and 423 spinal and 1032 epidural local anesthetic use for 'clinical' definition ( with a total of 2992 GA with 1455 NA?) BUT do not provide any explanation of data on conversion from neuraxial to GA rate or CSE use… ???

Response: We appreciate the Reviewer’s valuable comment. Our claims-based database does not distinguish between the conversion from failed neuraxial anesthesia to general anesthesia and the combined use of general and neuraxial anesthesia. Perhaps cases of L008 with NMBAs include conversion or concomitant use, but it is impossible to distinguish between the two. Nonetheless, in the L008 without NMBAs group, we suspect that neuraxial anesthesia was the primary anesthetic method in most cases and that the addition of any sedative was misclassified as general anesthesia.

In accordance with the reviewer's valuable suggestion, we have added the following explanation in the limitations paragraph of the Discussion (page 11, lines 294–298):

Our claims-based database does not distinguish between the conversion from failed neuraxial anesthesia to general anesthesia and the combined use of general and neuraxial anesthesia. Cases of L008 with NMBAs could include conversion or concomitant use; however, it is impossible to distinguish between the two retrospectively.

Comment 4: The authors do not provide data on the number of deliveries covered when reporting data for 15 years. They initially reported data from 2506 units; after excluding missing data records (1460), there are 1111 units that uploaded records. It is impossible to find out what was CS rate for this period.

Response: We appreciate the Reviewer’s valuable comment. In Japan, normal spontaneous vaginal delivery (NSVD) is not covered by health insurance; therefore, we cannot capture and estimate the number of NSVDs using the claim databases. In addition, the number of NSVDs it is not directly related to the scope of the current study. According to a governmental survey, the estimated number of NSVDs was approximately 1 million in 2014, and 18.6% of all deliveries were cesarean deliveries. In accordance with the Reviewer’s comment, we have added the following text to the Introduction (page 1, lines 32–34):

In the Japanese perinatal care delivery system, there are over 2,500 medical facilities for approximately 1 million deliveries per year, of which 18.6% were cesarean deliveries [1,2].

References

[1] Hasegawa, J.; Katsuragi, S.; Tanaka, H.; Kurasaki, A.; Nakamura, M.; Murakoshi, T.; Nakata, M.; Kanayama, N.; Sekizawa, A.; Isamu, I.; et al. Decline in Maternal Death Due to Obstetric Haemorrhage between 2010 and 2017 in Japan. Sci Rep 2019, 9, 11026, doi:10.1038/s41598-019-47378-z.

[2] Maeda, E.; Ishihara, O.; Tomio, J.; Miura, H.; Kobayashi, Y.; Terada, Y.; Murata, K.; Nomura, K. Cesarean Delivery Rates for Overall and Multiple Pregnancies in Japan: A Descriptive Study Using Nationwide Health Insurance Claims Data. J Obstet Gynaecol Res 2021, 47, 2099–2109, doi:10.1111/jog.14772.

Comment 5: Following that, if one considers provided data correct, it gives 10972 GA of all CS done in 1111 units for 15 years resulting in a performance rate of fewer than 10 CS per 15 years per unit = less than 1 GA cs per unit per year???

For clinical definition, even less … 2992 GA, which results in 2,69 GA CS for a unit for 15 years (?!?) = 0,18 of GA CS/unit/year, which is an extremely low number…. And indicates possible underreporting of GA use from very selected groups/units/patient population. Not consecutive cases allow for data generalization for the whole population of Japan…

Response: We appreciate the Reviewer’s valuable comment. Our database is an insurance-based database and is not institution-based (i.e., it does not include consecutive cases). Therefore, the number of cases at specific facilities were not accumulated and are relatively small. Compared to the healthcare system in Western countries, in Japan, small-scale obstetrical facilities provide delivery services through a decentralized delivery care system; there are 2,500 medical facilities for approximately 1 million deliveries per year, and 45.5% of all deliveries are performed in small-scale private obstetric facilities with <20 beds that are managed by a few obstetricians. These two reasons explain the issue raised by the reviewer.

We do not agree with the need for consecutive general anesthesia cases to generalize for the whole population of Japan. For example, consecutive sampling from university hospitals undertaking high-risk cesarean deliveries cases can bias generalizability. Instead, random sampling allows for overall data generalization. Our method samples the Japanese insured population under universal health coverage; thus, we believe that our method enables the description of the current status of general anesthesia practice in Japan. In view of the characteristics of the Japanese healthcare system, the inclusion of a diverse range of facilities would conversely ensure generalizability.

We have described this point as follows:

“The strength of our study is that it is the largest nationwide study to include diverse facilities and reflects the current state of clinical practice to date. The JMDC database is limited to the employee-based insured population; however, the Japanese health care system has universal health coverage. Hence, different types of health insurance or racial/social-economic disparities would not influence our results. Thus, our findings are likely indicative of the majority of cesarean deliveries performed in Japan [9].”

In accordance with the Reviewer’s comment, we have added the following text to the Introduction (page 1, lines 32–34):

In the Japanese perinatal care delivery system, there are over 2,500 medical facilities for approximately 1 million deliveries per year, of which 18.6% were cesarean deliveries [1,2]. Compared to Western countries, the Japanese delivery system is characterized by small-scale facilities that handle deliveries in a decentralized manner with localized networks.

We have added the following reference in the Reference

[1] Hasegawa, J.; Katsuragi, S.; Tanaka, H.; Kurasaki, A.; Nakamura, M.; Murakoshi, T.; Nakata, M.; Kanayama, N.; Sekizawa, A.; Isamu, I.; et al. Decline in Maternal Death Due to Obstetric Haemorrhage between 2010 and 2017 in Japan. Sci Rep 2019, 9, 11026, doi:10.1038/s41598-019-47378-z.

Minor Comment 1: Reported data are doubled in Figures and tables (Figure 2 and Table 3)

Response: We appreciate the Reviewer’s valuable comment. Accordingly, we have moved Table 3 to the Supplementary section to avoid duplication.

Minor Comment 2: Incorrect order of tables

Response: We appreciate the Reviewer’s comment and have corrected the order of the tables.

Minor Comment 3: The details of the database should be included in the manuscript's content, avoiding unnecessary citations.

Response: We appreciate the Reviewer’s valuable comment and have accordingly deleted the following text from the Methods.

The details of the JMDC database and included variables are described in previous health care researches [7,8,9].

Minor Comment 4: Avoid terms like "real-world clinical practice" 

Response: We appreciate the Reviewer’s valuable comment. We have revised terms like “real-world” throughout the manuscript.

Reviewer 3 Report

Presented article is very nice designed and written.

Please consider to add and discuss some recent articles regarding this topic (EJA 2019 - Changes in caesarean section... etc.)

Author Response

Response to the Reviewer #3

We wish to express our appreciation to the Reviewer #3 for providing insightful comments, which have helped us improve our manuscript significantly. We agree with all comments and have incorporated them into the R1 version of the manuscript.

Comment 1: Presented article is very nice designed and written. Please consider to add and discuss some recent articles regarding this topic (EJA 2019 - Changes in caesarean section... etc.)

Response: We appreciate the Reviewer’s valuable comment. Accordingly, we have added and discussed the following reference in the Discussion (page 10, lines 242–243):

“Czech and Slovak national surveys revealed an increase in the use of neuraxial anesthesia, but the proportion of general anesthesia remained high (24%) [22].”

[22] Stourac, P.; Kosinova, M.; Blaha, J.; Grochova, M.; Klozova, R.; Noskova, P.; Seidlova, D.; Richterova, S.; Firment, J.; OBAAMA-INT Study Group Changes in Caesarean Section Anaesthesia between 2011 and 2015: Czech and Slovak National Surveys. Eur J Anaesthesiol 2019, 36, 801–803, doi:10.1097/EJA.0000000000001063.